# Attempt to Extend the Shelf-Life of Fish Products by Means of Innovative Double-Layer Active Biodegradable Films

**DOI:** 10.3390/polym14091717

**Published:** 2022-04-22

**Authors:** Joanna Tkaczewska, Ewelina Jamróz, Paulina Guzik, Michał Kopeć

**Affiliations:** 1Department of Animal Product Technology, Faculty of Food Technology, University of Agriculture, Balicka Street 122, 30-149 Kraków, Poland; paulina.guzik@urk.edu.pl; 2Department of Chemistry, University of Agriculture, Balicka Street 122, 30-149 Kraków, Poland; ewelina.jamroz@urk.edu.pl; 3Department of Agricultural and Environmental Chemistry, University of Agriculture in Kraków, Mickiewicza Street 21, 31-120 Kraków, Poland; michal.kopec@urk.edu.pl

**Keywords:** double-layered films, furcellaran, active protein hydrolysates, lingonberry extract

## Abstract

In this study, we aimed to produce, innovative and, at the same time, environmentally-friendly, biopolymer double-layer films with fish processing waste and active lingonberry extract as additives. These double-layered films were based on furcellaran (FUR) (1st layer) and carboxymethyl cellulose (CMC) with a gelatin hydrolysate (HGEL) (2nd layer). The aim of the study was to assess their impact on the durability of perishable salmon fillets during storage, and to evaluate their degree of biodegradation. The fillets were analyzed for changes in microbiological quality (total microbial count, yeast and molds, and psychrotrophic bacteria), biogenic amine content (HPLC), and lipid oxidation (peroxidase and acid values, TBARS). The degree of biodegradation includes analysis of film and compost chemical composition solubility, respiratory activity, and ecotoxicity testing. The obtained results allow to suggest that active films are not only bacteriostatic, but even bactericidal when they used to coat fish fillets. Concerning the group of samples covered with the double-layer films, a 19.42% lower total bacteria count was noted compared to the control samples. Furthermore, it can be observed that the applied double-layer films have a potentially strong inhibitory effect on the accumulation of biogenic amines in fish, which is correlated with its antimicrobial effect (the total biogenic amine content for control samples totaled 263.51 mg/kg, while for the double-layer samples, their value equaled: 164.90 mg/kg). The achieved results indicate a high biodegradation potential, however, a too low pH of the film results in limiting seed germination and growth. Despite that, of these, double-layer films are a technology that has applicative potential.

## 1. Introduction

The main issue for future research is the search for new packaging materials having the desired functions and the development of new technologies that improve the carrier properties of these active biodegradable packaging types [1]. Furthermore, the current consumer trends such as more fresh and convenient foods, as well a desire for safer and better quality foods, impact packing technology of food products. Given these issues, consumers are demanding that food packaging materials be more natural, disposable, potentially biodegradable, as well as recyclable [2].

Proteins, polysaccharides, and lipids are the main biopolymers used in the production of biodegradable films and coatings. The enrichment of these coatings with active additives allows for the production of active packaging. These are systems that actively change the conditions of packaged food, causing, e.g., extension of shelf-life, and thus, its expiration date [3]. Furcellaran (FUR) is a polysaccharide extracted from the red algae *Furcellaria lumbricalis*, which is negatively charged. It is a linear polymer composed of units comprising a fragment of (1 → 3) β-d-galactopyranose with a sulphate group at C-4 and (1 → 4)-3.6-anhydro-α-d-galactopyranose [4]. Carboxymethyl cellulose (CMC) is a soluble and inexpensive polyanionic polysaccharide derivative of cellulose that has been employed as an emulsifying agent in the cosmetic and pharmaceutical industry [5].

Lipid oxidation in foods causes the creation of toxic substances as well as the degradation of polyunsaturated fatty acids. One of the directions in the development of active packaging is the modification of their surface with the use of carriers containing substances having antioxidant properties or the direct incorporation of such substances into the structure of the packaging material. High hopes are associated with substances of natural origin, as the use of synthetic antioxidants is more and more often of concern to consumers [4]. Appropriately selecting an active agent becomes a significant stage in manufacturing antioxidant and antimicrobial films used for food packaging. The active compounds and the film-forming material should be compatible, and thus, promote uniform release of the active ingredient over the entire surface of the packed food. In the last decade, many edible coatings containing antioxidants and antimicrobial agents have been tested as potential active packaging materials [5,6]. The possibility of using protein hydrolysates and various plant extracts as active ingredients was also investigated.

Plant extracts are often used in edible films because they are deemed safe when used in low concentrations. Synthetic and natural food-preserving agents may be incorporated, however, natural plant extracts are, in the majority of cases, preferred in comparison to synthetic alternatives. Plant extracts use implemented in films comprise those from dried flowers, fruit, leaves, and agricultural waste via extraction technologies. Phenolic plant extracts, having high amounts of phenolic acids, flavonoids, anthocyanins, tannins, as well as other polyphenol sub-groups, are rich in dietary antioxidants [6]. The Vaccinium species—lingonberry (*Vaccinium vitis-idaea* L.) in particular—is constantly being noted as plant-derived functional products due to their diversity in phenolic compounds. In recent literature, it has been described that lingonberry occupies a significant place on the antioxidant and antimicrobial capacity ranking with regard to Vaccinium-derived species [7].

Fish protein hydrolysates (FPH) having biological properties are also a frequent subject being of particular interest among pharmaceutical, functional food and food processing/preservation industries. Obtaining protein hydrolysates from food industry waste materials, which are the source of antioxidant peptides, can be a way to manage them and obtain relatively inexpensive antioxidant preparations [8]. It has been previously noted that Carp skin gelatin hydrolysates (CSGH), from carp industry waste, demonstrate potent antioxidant activity [9]. The hydrolysate’s antioxidant activity is basically caused by the presence of dipeptide Alanine-Tyrosine (Ala-Tyr) [10]. The coating made with the carp skin gelatin hydrolysate has the potential to extend the shelf-life of packed products, while additionally causing the conversion of food industry by-products into value-added, film-forming components [11].

Marine foods rapidly deteriorate post-mortem as a consequence of various biochemical and microbial breakdown mechanisms [12]. Due to the growing demand for high-quality, fresh-fish products, a lot of research is being focused on finding methods to extend the shelf-life of fish. One such strategy is the use of coatings containing antimicrobial and/or antioxidant compounds that, when applied to the product, have a preservative effect [13].

In our previous research, we have demonstrated a method for achieving new biopolymer double-layered films on the basis of furcellaran (1st layer) and carboxymethyl cellulose, combined with a carp skin gelatin hydrolyzate (2nd layer) [14]. Due to the presence of active ingredients in the lingberry extract, an increase in antimicrobial activity against Gram-positive and -negative bacteria was observed in the films. Moreover, CSGH addition caused the films to exhibit antioxidant activity as measured via FRAP and DPPH tests. The obtained films were characterized by very favorable mechanical properties and high biological activity. In another of our studies [15], a double-layered film with CSGH and lingberry extract was part of the whole packaging during salmon storage. We discovered that this total packing shows a strong inhibitory effect on the accretion of bioamines, while inhibiting the growth of bacteria responsible for the spoilage of fish products.

As we wanted to provide a more precise characterization of the active films based on CMC and furcelleran, we carried out another experiment to assess the bio-degradation degree of the obtained films. This allowed to further evaluate their potential impact on the environment. Another goal of the study was to assess the possibility of using these films as an active packaging for fish products, in order to extend their shelf-life. The study on the impact of these double-layer films on the quality of the packaged product during storage is necessary for their proper commercialization as an innovative, biodegradable active packaging.

## 2. Materials and Methods

### 2.1. Materials

The authors obtained the furcellaran (FUR) (type 7000) used in the study from Est-Agar AS (Karla village, Rae Parish, Estonia). The chemical content of the FUR chemical (M_w_ 2.951 × 10^5^) included the following: 79.61% carbohydrates, 1.18% of protein, and 0.24% fat. Carboxymethyl cellulose (CAS:9004-32-4) was procured from POL-AURA (Zabrze, Poland). We acquired the lingonberry fruits from the Natura Wita company (Pińczów, Poland).

Salmon (*Salmo salar*) fillets were purchased from a fish processor (Sona, Koziegłowy, Poland) to conduct the experiments on fish preservation.

### 2.2. Methods

#### 2.2.1. Film Preparation

Common carp (*Cyprinus carpio*) skins were obtained from a local carp processor (Sona, Koziegłowy, Poland). The tested carp skins were ground (MADO MEW 613 grinder (Dronhan, Germany) after removing the remaining tissue. We procured the carp skin gelatin and hydrolysates according to the method suggested by [9,16].

The procedure for obtaining the double-layered films was discussed in our previous work (Jamróz et al., 2021). A lingonberry extract solution was prepared. The lingonberry fruits were blended (Model 00753476, Gerlingen, Germany). Then, 20 g of the lingonberry powder was weighed, to which 200 mL of H_2_O was added. The prepared solution was stirred at 70 °C for 30 min. The extract was put through a filter to obtain a clear solution. The extract was characterized by the ability to scavenge DPPH free radicals at the level of 42.51% and the total polyphenols content 4524.18 gallic acid mg/L.

Using carboxymethyl cellulose (CMC—by adding 1% *w*/*v*, 1 wt% glycerol following dissolution—the solution was stirred (MR, Heidolph, Schwabach, Germany) for 12 h at room temperature) and furcellaran (FUR—1% *w*/*v*, 1 wt% glycerol was added post-dissolution—the solution was then heated to 130 °C for 20 min), the individual solutions were prepared.

The lingonberry extract was mixed with the FUR solution to obtain an ultimate 40% *v*/*w* concentration. Following this step, the mixture was poured onto rectangular polypropylene forms (30 cm × 20 cm). This was performed until a gel was achieved. To 1% *v*/*v* CMC, 0.5 g of the gelatin hydrolysate (HGEL) was added. When the solution turned clear, it was pipetted onto the 1st FUR layer. The film-forming solutions were then dried in room temperature conditions under a frame hood. After becoming completely dry, we peeled the solutions from the forms to carry out consecutive tests.

#### 2.2.2. Biodegradation Assessment of Films

##### Film and Compost Chemical Composition

In order to determine chemical composition, the samples taken from initial materials and composts were subjected to incineration in a chamber furnace (Czylok, Jastrzębie-Zdrój, Poland) at a temperature of 800 °C for 12 h. Following digestion of the ash in diluted (1:2) HNO_3_, the samples were transferred to flasks. Next, by adding redistilled water, the volume of the final sample was adjusted to 50 mL. The content of the examined elements (in the obtained solutions) was determined via the ICP-OES procedure. The tests were performed in duplicate.

The pH of the compost suspended in water (at a ratio of 1:10) was determined potentiometrically (pH-meter CP-505, Elmentron, Zabrze, Poland). This was performed analogously in the case of electrical conductivity (EC) by implementing the CCO-501 conductometer (Elmentron, Zabrze, Poland). The content of total carbon and nitrogen were established using the Vario MAX Cube analyzer, which is equipped with an IR sensor (Vario MAX Cube, ElementarAnalysensysteme, GmbH, Langenselbold, Germany). In order to determine the content of ash, the samples taken from the baseline materials were also subjected to incineration in a chamber furnace at a temperature of 800 °C for 12 h. The tests were performed in duplicate.

##### Solubility of the Tested Films

The films were cut into 3 cm × 3 cm squares, which were then weighed to the nearest ~0.0001 g (W_1_—initial weight). After weighing, the films were dried in an oven at 70 °C for a period of 24 h in order to achieve the initial dry matter (W_2_). Then, the samples were put into 30 mL of Milli-Q water for a duration of 24 h. Afterwards, we removed the film samples. The film samples were once more dried in an oven at a temperature 70 °C for a 24-h period to obtain their non-dissolved, ultimate dry weight (W_3_). Following the drying procedure, they were weighed. In accordance with the standard equations given below, the following were assessed: water content, film solubility, and swelling degree. The tests were performed in triplicate.

##### Degree of Biodegradation

In order to test the degree of biodegradation, we subjected the double-layered films to incubation based on 14855-1:2005. The respiratory activity of the material during stimulated biological changes (composting) was determined in controlled composting conditions, taking the evaluated losses in dry matter as well as their rate into account. The samples totaling 10 g of the compost fresh weight, including 1 g of the film’s dry weight, were used for testing. They were subjected to incubation (Czylok, Jastrzębie-Zdrój, Polska) for a period of 14 days. Furthermore, 58 ± 1 °C was the temperature chosen for use in the incubation of these films with an applied medium. 

Vermicompost was taken as a substrate after 5 months of its transformation with *Eisenia fetida*, implementing a 50% moisture rate. The incubation of the double-layered films conducted in vessels with a 2.5 dm^3^ capacity. After completing the composting process, samples were taken from the tested materials. Moreover, the content of their dry matter was determined via a moisture analyzer, the program set to the temperature of 105 °C, and the reaction accuracy being 0.001 g/60 s. The discussed analyses were performed in duplicate.

##### Respiratory Activity of the Material during Stimulated Biological Changes (Composting)

To measure respiratory activity during stimulated biological changes (composting) (influence of changes in CO_2_ for closed-vessel volume) with regard to the incubated double-layered films with the compost, the manometric method was applied via Oxi-Top measuring equipment. The manometric measurement of respiration activity concerning the analyzed materials included registering changes within the pressure of the closed-vessels being in a continuous system. The recorded pressure-related changes directly proportional to the amount of oxygen in-took by the sample, which was the result of breathing processes that automatically occur in the sample (OxiTop^®^ 2003). These pressure changes were noted every 60 min. The amounts of the resultant CO_2_ equivalent were then absorbed by the 1-mol × dm^−3^ NaOH solution which was within the vessels. The system used for measuring the respiration activity consisted of measuring bottles and applicable accessories. At the time of determination, the measuring bottles were placed in a thermostatic cabinet, having a constant temperature totaling 58 ± 0.1 °C. Using an infrared interface, the acquired data were transferred to the controller, and later further transferred to a computer by implementing the Achat OC program. The respiratory activity of the materials was converted to dry weight. The standard formula (1) was applied for calculations. The tests were performed in triplicate.
RA = (MCO_2_/RxT) × (Vfr/mBt × |Δp| [mg CO_2_ × (g × h)^−1^](1)
where: RA—respiratory activity; MCO_2_—molar mass of CO_2_ (44,010 mg. Mol^−1^); R—general gas constant (83.14 L hPa. (K. Mol^−1^)^−1^; T—temperature of measurement (K); mBt—composted material dry weight (kg); | Δp |—pressure changes (hPa) and Vfr—free gas volume.

##### Ecotoxicity Testing

In closed-vessel conditions, analysis of cress growth (*Lepidium sativum* L.), in contact with procuring the extracts (raw material ratio: water 1:50, extraction 24 h), was performed in tandem with measurements of carbon dioxide absorption and manometrics (OxiTop^®^ measuring system, Wageningen University and NMI, 13 pp. (2003)) (Kopeć, Baran, Mierzwa-Hersztek, Gondek, & Chmiel, 2018). The control object had a vessel; in it, testing for seedling growth test was carried out exclusively against redistilled water. In the evaluation, seed weight (0.268 ± 0.002 g) was considered. Formula (1) was used to obtain the results of seedling growth testing. The degree of toxicity was estimated as a percentage of the resultant CO_2_ with respect to the object placed in distilled water. These analyses were carried out in duplicate.

#### 2.2.3. Fish Preservation Testing

##### Preparation of the Sample

The innovative films were place on raw salmon fillets in order to determine efficiency. Three filets, selected at random, were used for each day of testing. The fish samples were divided into 2 groups: (1) those wrapped in synthetic films—the control group (LDPE), and (2) those wrapped in biopolymer films with added lingonberry extract (FILMS group).

The samples were placed on PET trays and stored at 4 °C. Following the 0, 3, 6, and 10 days of storage, the films were peeled off from the fish fillets and the samples were homogenized (R2 Robot Coupe, Vincennes, France). Then, they were subjected to further test.

##### Analyzing Microbiological Properties

A 10-g total was taken from each fish sample and further mixed with 90 mL of maximum recovery diluents (Oxoid, Basingstoke, UK). This was homogenized for 120 s in a Stomacher blender. Total viable count (TVC) analysis was carried out on the basis of the pour plate method, which was performed on a Plate Count Agar (Biomaxima, Warsaw, Poland), with the plates incubated for 48 h at 30 °C. Yeast and mold (YM) count was performed via the spread plate method on a DRBC agar (Biomaxima, Warsaw, Poland). The plates were incubated for 120 h at a temperature of 25 °C. Analysis of psychrotrophic bacteria count was performed on a spread plate with the Plate Count Agar (Biomaxima, Warsaw, Poland). The plates were incubated for 240 h at 6.5 °C.

##### Analysis of Biogenic Amines

The analysis of biogenic amine was performed through the implementation of the HPLC method, including dansyl chloride derivatisation. The samples were prepared according to the methods proposed by Kulawik, Dordevic, Gambuś, Szczurowska and Zając (2018). Sample separation was performed using the Dionex Ultimate 3000 UHPLC (Thermo Scientific, Waltham, MA, USA), FLD 3400RS four-channel fluorescent detector (Thermo Scientific) on a Kromasil 100-5-C18 4.6 × 250 mm column (Akzo Nobel, Amsterdam, The Netherlands). In an earlier study by Jamróz, Kulawik, Guzik, and Duda (2019), specific chromatographic conditions were presenting. The Supelco biogenic amine standards were used for referencing (Sigma-Aldrich, St. Louis, MO, USA).

##### Oxidation Rate of Fish Lipids

The acid value of the samples was using the method of titration (AOCS Cd 3d-63, 1997). The value of peroxidate was established via the Acetic Acid-Chloroform procedure (AOCS Cd 8-53, 2003). TBARS analysis was carried out as previously defined in [15].

#### 2.2.4. Statistical Analysis

Statistical analyzes were performed by implementing the NCSS 2020 statistical package. For biogenic amines, two-factor analysis of variance was applied. The type of packaging was considered as the grouping factor, while the date was assumed as the repeated measure factor. Post-hoc analysis of contrasts was carried out by comparing the results obtained for both groups and for each term. With regard to the results achieved for peroxide and acid numbers, the packages were compared with the Mann–Whitney U test. The adopted level of statistical significance was *p* < 0.05, while *p* < 0.01 was assumed as highly significant.

All in vivo (with fish samples) analyses were performed using 3 independent repetitions.

## 3. Results and Discussion

### 3.1. Results of Biodegradation

The obtained films were characterized by a low pH value (3.2), which results from the presence of a high concentration of lingonberry extract at a pH totaling 3.0. The introduction of material into the compost improves environmental conditions for developing biodegradable microorganisms. Organoleptically, no film residues were found in the incubated material after 14 days. After the end of incubation, the detected film loss indicated greater dynamics (the loss of mass was 84.95%), which is related to the high solubility of this material (approx. 72.79%) (Table 1). The biodegradability of films may also be affected by the biopolymers’ water absorption ability. Microorganisms employ a polymer matrix as a source of energy, while the water that has been absorbed facilitates microorganism development on the film. This further supports the enzymatic activity of microorganisms weakening polymer chains. Thus, degradation of the material occurs faster in cases where the amount of water diffused via the films is greater [17]. The addition of lingonberry extract to the double-layer films caused a decrease in WCA value [14] which, in turn, results in a more hydrophobic nature of the film, also translating into an extended biodegradation process. Wongphan, et al. [18] noted that high water absorption could accelerate the hydrolysis activity catalyzing the degradation and breakage of polymer bonds, including those glycosidic in thermoplastic starch (TPS) and polybutylene adipate terephthalate (PBAT) films.

The analysis of data on the course of respiratory activity indicates high dynamics of microbiological changes (Table 2).

With regard to respiratory activity, on the 1st day of incubation, its level was already noted as high, then, it experienced a decrease after 4 days (96 h later). The equations concerning respiration activity at the time of composting indicate varying dynamics at the facilities. On the basis of the directional coefficients of the linear regression equation, the conclusion was reached that the films supplemented in the vermicompost caused 3.7-, 2.3-, and 1.4-times greater respiratory activity in comparison to the control. These results allow to suggest very high microorganism activity which decomposed the added substrates on the 1st day of incubation, while quenching of the process became similar as the time passed.

The films based on polysaccharides obtained from algae (furcellaran) biodegraded in a very short time, which is related to the presence of α-1,3 and β-1,4-glycosidic linkages between monomeric units. This type of weak linkage is easily disrupted by various microorganisms, with the consequent formation of short chain polysaccharides, which are then incorporated into metabolic pathways when further broken down [19]. CMC-based films are completely biodegradable within 7 days [20], this being related to their hydrophilic nature. In addition, hydrophilic cellulose derivatives easily absorb water from the environment which, in turn, has impact on the plasticization of the material and its swelling [21]. In the case of furcellaran and CMC film solubility, which is 100%, this is similar. The obtained double-layered films not only have reduced solubility, but also due to the prepared system, they are biodegradable for a longer period of time. The addition of lingonberry extract made the films more hydrophobic [14]. This behavior may be related to the development of interactions between functional groups among furcellaran and the functional groups of the lingonberry extract. Furthermore, there may be stronger interactions (for example, hydrogen bonding) between the 1st and 2nd layers of the films. Wongphan et al. [22] reached similar conclusions in the case of edible starch films incorporating papain.

Biological tests were carried out to determine the effects of the double-layered films on the germination capacity of *Lepidium sativum* L. seeds (Figure 1). It was observed that kernel growth was slower after the addition of the film with lingonberry extract than in the control sample. From the calculated amount of CO_2_ formed during the 4 days of seedling growth, it was found that the extract scabbed in the films developed 61.4% worse in relation to the area with distilled water. The limitation of cress germination and growth is related to the low pH (3.21) value of the film.

The obtained results indicate high biodegradation potential, however, the too low pH of the film results in limiting seed germination and growth.

### 3.2. Fish Preservation Testing

The next step of the experiment was to use the resulting films as single packing materials for the salmon (Figure 2).

#### 3.2.1. Microbiological Analysis

Microbial population changes, with regard to salmon (*Salmo salar*) fillets wrapped in bioactive films, are demonstrated in Figure 3 for the storage period.

The total number of microorganisms (TVC) for the control samples experienced a regular increase as the storage time progressed, accomplishing the highest values towards the end of the period devoted to chilling (5.75 colony-forming unit/g-8.75 colony-forming unit/g). A completely different relationship was observed for the samples of salmon stored in active films. After 3 days of storage, there was a reduction in the bacterial population by 0.72 log cycles (4.98 log CFU/g) compared to day 0 (5.75 log CFU/g). The bacterial population on day 6 of storing the samples increased to 7.37 log CFU/g, but on day 10, it slightly decreased once more (7.05 log CFU/g). On this basis, it can be assumed that the active films not only have a bacteriostatic, but even bactericidal effect.

In the authors’ previous research, it was proved that the analyzed films showed very good in vitro antimicrobial effects against strains of *Escherichia coli*, *Enterococcus faecalis*, *Staphylococcus aureus*, *Salmonella enterica*, and *Pseudomonas aeruginosa* [14]. The obtained results allowed us to confirm that the designed active films can also be effective in preventing the development of microorganisms in charge of the spoilage of food products in vivo. The inhibitory effect of lingonberry extract on microorganisms present in salmon fillets may result from the action of hydrolyzable tannins (ellagitannins), anthocyanins, and benzoic acid. The recommended mechanisms that allow to explain the antimicrobial activity of tannin comprise extracellular microbial enzyme inhibition, deprivation of substrates needed for microbial growth, or direct action regarding microbial metabolism via inhibition concerning oxidative phosphorylation or iron deprivation [23].

The inhibitory effects on the growth of microorganisms within the product could be caused by the presence of the CSGH in the films, since films with hydrolysates demonstrate increased oxygen diffusion barrier properties, therefore, inhibiting bacterial proliferation through forming a protein bio-film around the fish samples [24]. Furthermore, gelatin hydrolysates are well-known for their antibacterial effects against concrete chilled fish spoilage organisms, among others, *Shewanella putrefaciens* and *Photobacterium phosphoreum* [25]. Likewise, Jridi, Mora, Souissi, Aristoy, Nasri, and Toldrá [26] have presented antibacterial effects of fish gelatin coatings on protecting meat against bacterial spoilage at the time of cold storage. The peptide antimicrobial activity, and therefore, also, the protein hydrolysates containing them, significantly differs according to the amino acid profile, structure as well as peptide length and sequence [27]. The mechanism of antimicrobial peptide action is chiefly based on the peptides’ electrostatic interaction with the cell membrane of microorganisms. Antimicrobial peptides demonstrate membrane permeabilizing properties. They are able to enter the membrane, which results in its disruption [28]. In accordance with the work by Verma, Chatli, Mehta, and Kumar [29], the bioactive protein hydrolysates antimicrobial activity could additional be caused by iron chelating activity.

Throughout the whole period analyzed, the yeast and mold counts experienced an increase in both of the groups under evaluation. Nonetheless, in comparison to samples that were not coated, applying bioactive films on the surface of salmon caused yeast and mold growth to be inhibited during the 1st period of storage. Furthermore, the bioactive films caused the yeast and mold growth to be inhibited on the 3rd day of storing the fish in comparison to the control samples (by 0.69 log cycles). Starting from the 6th day of storage, there were no differences noted with regard to the amount of yeast and mold found in the case of samples from the control or study groups. In the authors’ previous research, it was proved that the designed innovative films showed in vitro activity, inhibiting the growth of *Candida albicans*, *Candida krusei*, *Aspergillus brasiliensis*, and *Aspergillus flavus* yeast and molds [14]. However, the obtained results indicate that the tested films slightly prevent the growth of yeasts and molds during the 1st period of storage if they are applied to the surfaces of perishable products. On the following days of storage, the bioactive films do not show any antifungal properties. A possible explanation for this phenomenon may be that the lingonberry extract does not contain ellagitannins responsible for the antifungal and anti-mold effects [7]. The same authors [7] have reported that lingonberry extract shows little antifungal properties. The minimum inhibitory concentration for Candida albicans was 125 mg/mL and the minimum fungicidal concentration was 250 mg/mL. Furthermore, according to Özvural et al. [30], encapsulation and binding plant extracts in the coating may prevent its antifungal activity. What is more, the release rates and behavior of antimicrobial substances depend on several factors, including polymer types, the method and process of film preparation, film microstructures, antimicrobial-polymer interactions, and environmental as well as medium conditions. In vitro antimicrobial testing with several microbial strains and real packaged foods provided diverse results because release mechanisms depend on food components and testing conditions [31].

A similar study was conducted by Tumbarski et al. [32]. The authors examined the effects of 1% carboxymethyl cellulose edible films containing 3 different ethanolic propolis extracts on microbial quality of cheese. The application of the CMC edible films with propolis ethanolic extracts on kashkaval cheese did not influence the total yeast count, and in all treatments, until the end of the storage period, the yeasts remained within ranges close to those of the control samples.

When aerobically storing fresh fish at chilled temperatures, psychrotrophic bacteria are the microorganism group causing spoilage [33]. The growth pattern of psychrotrophic bacteria demonstrated the same behaviors as those of TVC. As time passed, the psychrotrophic bacteria count increased from 4.96–8.80 log cfu/g in the control samples. On the 3rd day of storage, the samples covered with the active films had a lower number of psychrophilic bacteria by 2.08 log cycles compared to the control sample. This trend remained throughout the whole period devoted to storage, the difference totaling 1.72 log cycles which was detected on the 10th day of storage.

According to the HPA (2009) [34], the limit established for bacterial counts is 10^6^–10^7^ CFU/g, Therefore, the shelf-life of samples from the control group would only be 3 days, while for samples from the bioactive films group, this shelf-life could be extended over the next 3 days.

#### 3.2.2. Content of Biogenic Amines in Fish Samples

The following biogenic amines were determined: β-phenylalanine, cadaverine, histamine, putrescine, tryptamine, tyramine, spermidine, and spermine (Table 3). The storage periods of *Salmo salar* were prolonged well beyond the accepted limit.

The amines formed in salmon fillets showed a high baseline amount for both study groups. Comparing the total amount of biogenic amines from the control sample at the end of the storage period to the number amount of biogenic amines from the coated samples, it may be noted that the implemented double-layer films demonstrate a potential, strong inhibitory effect with regard to biogenic amine accumulation (the value for control samples totaled 263.51 mg/kg, while for the double-layer samples, their value equaled: 164.90 mg/kg).

Cadaverine and putrescine are converted from lysine and ornithine via action of corresponding bacterial decarboxylases. This causes them to be major contributors to the perceived off-flavor of spoliating fish [35]. In this trial, the baseline cadaverine and putrescine concentrations equaled 1.46 and 3.50 mg/kg, respectively. Both values maintained moderate levels during the early-on storage period. An acute increase in the level of cadaverine could be seen up until the 3^rd^ day, both in the control and double-layer groups. This increase was further in accordance with the abrupt rise in TVC. The application of double-layer films considerably caused the accumulation of putrescine and cadaverine to be inhibited (*p* < 0.01). A highly significant (*p* < 0.01) difference also applied to the interaction—the increase in cadaverine concentration for the samples covered with the control film is much more dynamic than in the study group. The inhibitory effect of double-layer films on biogenic amine formation was mainly due to the antimicrobial effect of lingberry extract. It has been reported that putrescine accumulation has a close relationship with the activities of *Pseudomonas* bacteria spoilage, while *Aeromonas* and *Shewanella* are more responsible for cadaverine production [35]. According to Chatkitanan and Harnkarnsujarit [36], microbial growth could cause modification of protein conformation, including secondary structures and proteolysis which modify amino acid profiles in muscle foods. It could affect the creation of biogenic amines in salmon samples.

Histamine is a causative agent inducing fish spoilage. A highly significant difference in the content of histamine was found between the study groups since day 6 of storage. The samples covered with the innovative films contained significantly less of this amine compared to the controls. According to the CAS, fish and their products should not contain more than 100 mg/kg of histamine as a hygiene limit [37]. The limit for histamine was exceeded in the fish from the control group on the 10th day of storage (160.92 mg/kg), while for fish covered with innovative films, the level was significantly lower on that day (87.44 mg/kg), not exceeding the limits. Therefore, it can be concluded that the use of innovative films extended the shelf-life of salmon fillets.

In the case of spermidine, one of the highest baseline concentration levels was noted among all of the studied biogenic amines. This is due to it being a natural substance of living cells [37]. During storage, in the control, and at the same time, in the study groups, a slow increase, or in some cases, even decreases, were observed in the trend changes related to spermidine and spermine. It was observed that the content of spemidine and spermine was significantly lower than in the control group. The interaction was also highly significant (*p* < 0.01)—in both groups, the content of spermine and spermidine decreased on day 3 and then increased, while in the control group, the increase was more dynamic.

The content of phenylethylamine, tyramine and tryptamine in the meat of the tested fish increased during storage in both test groups, but in the samples not covered with films, this amount was significantly higher. Moreover, compared to the study group, in the control group, a significantly more dynamic increase in the concentration of these biogenic amines was noted during storage. Compared with phenylalanine found in fish (>8.0 mg/kg) [38], its level in the study group was much lower (<4.26 mg/kg). Tyramine level in sample cover with films (<18.07 mg/kg) was much lower than it in carp (>23.2 mg/kg) [38]. Tryptamine in the experimental group was low content ranging from 0.17 to 1.26 mg/kg. Similarly, tryptamine found in chill stored red drum was of low concentration (<1.12 mg/kg) [39].

In the authors’ previous research, the use of active, single-layer biopolymer films based on furcellaran and gelatin hydrolysates from carp skins on the surface of Atlantic mackerel carcass did not inhibit the formation of biogenic amines in samples stored in refrigerated or freezer conditions [11,40]. Moreover, it was found that in the case of some biogenic amines, the use of films even increased their formation in fish samples, which may be due to the high content of free amino acids in the hydrolysate of carp gelatine, which are precursors of biogenic amines. New, innovative, double-layer biopolymer films were created as a result of complexed carboxymethyl cellulose and gelatin hydrolysates from carp skins, in the form of a single layer, on which a second layer containing furcellaran and lingberry extract was applied. This solution turned out to be effective in inhibiting the development of biogenic amine formation in fish samples. Consequently, this double-layer films application is a technology having the potential to be used for increasing the shelf-life of seafood.

#### 3.2.3. Lipid Oxidation of Salmon Fillets

The peroxide value (PV) provides a measure of the lipid oxidation degree and indicates the amount of oxidized substances [41]. When fats turn rancid, triglyceride is converted into fatty acid and glycerol and thus, increase the acid value (AV) of the samples. An increase in AV is generally associated with lipase activity originating from microorganisms or biological tissue [42].

The PV and AV in the salmon samples cover by different films are shown in Table 4.

The statistical analysis showed no influence of the used film on the PV and AV of salmon meat. Although there were no statistically significant differences between the films and LDPE groups, a positive effect of the biopolymer films on the extension of the tested samples’ durability could be observed. On the beginning of the test, acid value was 1.91 mg KOH/g (Table 2). On the final days of storage, acid value increased to 8.33 mg KOH/g and 4.05 mg KOH/g for the for control and study groups, respectively. The acceptable limit for acid value is reported to be 7–8 mg KOH/g [42]. Covering filets with a biopolymer films restricted the increase in acid value after 10 days of storage, and caused an increase in the shelf-life of the product.

According to the authors’ research, active films are characterized by high antioxidant properties measured via the FRAP method (41.89 mM Trolox/mg) [14]. Therefore, it was supposed that they would slow down the lipid oxidation process. The use of films causes there to be a protective film between the fish and the direct external environment. This is especially useful in the case of oxygen most closely linked with lipid oxidation. Biopolymer coatings or films can form as a barrier to oxygen as well as carbon dioxide [43]. Furthermore, according to Chatkitanan and Harnkarnsujarit [44], incorporating active ingredients containing hydrophilic groups into the films possibly reduced oxygen permeability, leading to lower oxygen transfer and lipid oxidation. The lack of statistically significant differences between the examined groups can be explained by the fact that the control sample was covered with a synthetic film, which also constitutes a barrier to oxygen and other gases.

According to results of TBARS analysis, covering fillets with the active films did not only inhibit lipid oxidation in salmon fillets when compared to the sample with LDPE films, but it even increased the process. The concentration of TBARS values in the film’s samples increased from 0.23 to 1.79, and from 0.23 to 0.34 mg malonaldehyde/kg in the LDPE sample during 10 days of storage (Table 2). The possible explanation for the higher value of TBARS in the film samples is the occurrence in its of polyphenols from lingonberry extracts [14]. Oussalah et al. [45] claims that the degradation pathway of some phenolic compounds can cause the formation of phenolic aldehydes. These substances may generate a similar reaction to the MDA reaction when lipid oxidation of the fish was assessed by the determination of TBARS. This could further result in increased TBARS values, and consequently, mask of the antioxidant effect of innovative films.

Some data from literature on the subject allow to confirm the effectiveness of inhibiting oxidation in fish products with the use of coatings containing antioxidant ingredients. For example, in a study by Sun et al. (2019) [43], it was observed that fish gelatin-based coatings containing curcumin slow down the processes of lipid oxidation in grass carp fillets. There are also reports in which no in vitro influence of coatings with high antioxidant potential on the inhibition of oxidation in model food products was noted. Oussalah et al. (2004) [45] created protein-based films enriched by adding pimento or oregano essential oils. In any case, the antioxidant potential of such films notwithstanding, the researchers did not record any protective effect against oxidation in the model meat product, which is in accordance with the findings reported in the present study.

## 4. Conclusions

Innovative, double-layer films based on carboxymethylcellulose and furcellaran enriched with active ingredients, i.e., lingonberry extract and gelatin hydrolysates from carp skins, are effective in inhibiting the growth of microorganisms responsible for the spoilage of a model food product (*Salmon fillet*). Moreover, the use of these double-layer films as active packaging materials has a strong inhibitory effect on the formation of biogenic amines in packaged fish filets. The confirmed toxicity of the active double-layered material, which is related to the low pH value of the films, may also inhibit the growth of microorganisms during the storage of the fish. The double-layer films are easily biodegradable and extend the shelf-life of salmon fillets, thus, demonstrating the environmentally-friendly potential as packaging for food products with a short shelf-life. For consumers, an increasing preference for foods that are considered high-quality, safe, as well as clean-labeled, can be noted. Therefore, it is probable that applying natural and ecologically-friendly films may become a future trend.

## Figures and Tables

**Figure 1 polymers-14-01717-f001:**
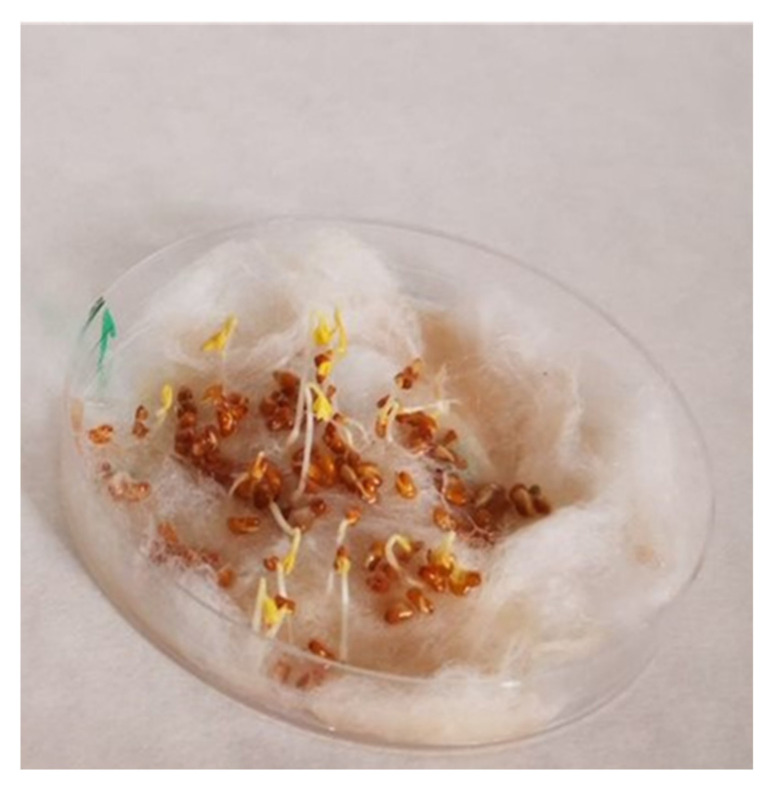
Photos of germination and growth of cress (*Lepidium sativum* L.) seeds with double-layered films.

**Figure 2 polymers-14-01717-f002:**
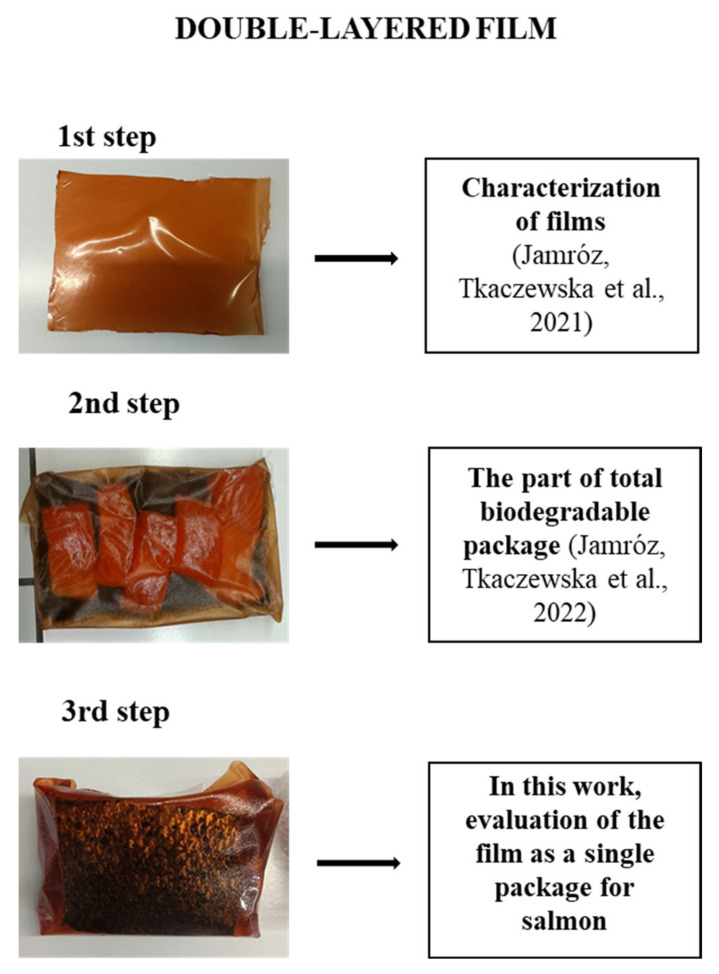
The use of double-layer films as packaging materials [14,15].

**Figure 3 polymers-14-01717-f003:**
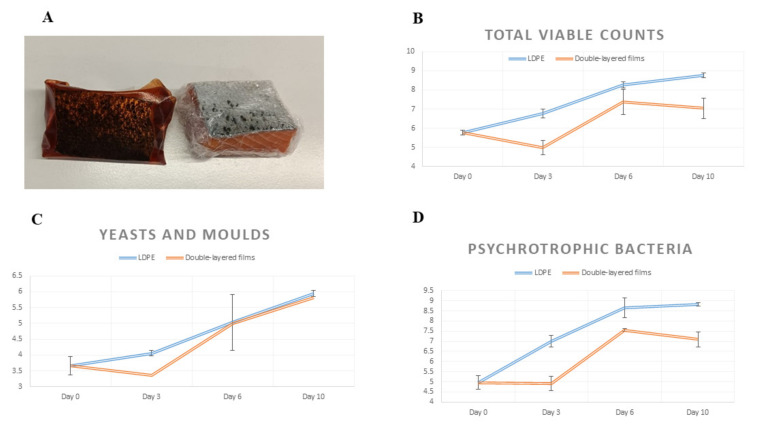
(**A**) The method of packing salmon in 2 types of films. (**B**) Total aerobic bacteria in *Salmo salar* fillets; (**C**) yeasts and molds in *Salmo salar* fillets; (**D**) psychrotrophic bacteria in *Salmo salar* fillets. Films: samples stored in innovative films; LDPE: control group samples stored in synthetic films.

**Table 1 polymers-14-01717-t001:** Parameters of incubated materials.

Parameter	Compost	Double-Layered Films
pH	8.12 ± 0.21	3.21 ± 0.25
EC [mS/cm]	2.88 ± 0.15	0.963 ± 0.05
N [g/kg]	2.443 ± 0.13	0.896 ± 0.04
C [g/kg]	30.65 ± 0.93	38.75 ± 0.99
C:N	12.54 ± 0.52	43.40 ± 1.50
Solubility [%]	-	72.79 ± 2.97
Degree of dry matter loss after incubation [%]	7.36 ± 5.01	84.95 ± 3.12 **
Ash [%]	-	4.95 ± 1.81

** after taking losses at the control facility into account.

**Table 2 polymers-14-01717-t002:** Equations of respiratory activity for incubated materials over 3 equal periods covering 14 days (mg CO_2_ ÷ g h) *.

	Period		
	0–112 h	113–224 h	224–336 h
Control	Y = 0.6512 + 8.7823 R^2^ = 0.9681	Y = 0.3023x + 41.982 R^2^ = 0.9794	Y = 0.248x + 53.545 R^2^ = 0.9834
Double-layered films	Y = 2.4205x + 7.7248 R^2^ = 0.9857	Y = 0.6976x + 173.71 R^2^ = 0.9866	Y = 0.3415x + 245.54 R^2^ = 0.9723

* based on the dry weight of the substrate. CONTROL: samples without films and the control object comprised a vessel. Double-layered films: the object with double-layered films.

**Table 3 polymers-14-01717-t003:** Biogenic amine concentration in fish samples through storage in 4 °C (mg/kg).

Group/Day		TRPYP	PHEN	PUTRY	CADAWE	HISTAM	TYRAM	SPER	SPR
**Initial (day zero)**		0.30 ± 0.07	0.68 ± 0.16	3.50 ± 0.68	1.46 ± 0.56	6.26 ± 1.175	2.34 ± 0.50	2.34 ± 0.50	4.01 ± 0.65
**LDPE**	**3**	1.08 ^b^ ± 0.12	0.62 ^b^ ± 0.03	2.83 ^b^ ± 0.40	29.83 ^b^ ± 7.67	26.36 ^a^ ± 4.17	3.91 ^b^ ± 1.09	2.12 ^b^ ± 0.06	2.94 ^a^ ± 0.34
	**6**	3.16 ^b^ ± 0.19	5.42 ^b^ ± 0.36	9.00 ^b^ ± 1.00	61.06 ^a^ ± 2.24	86.48 ^b^ ± 13.50	23.34 ^b^ ± 1.27	2.08 ^a^ ± 0.23	3.42 ^b^ ± 0.21
	**10**	3.73 ^b^ ± 0.34	10.26 ^b^ ± 2.09	11.86 ^b^ ± 1.28	43.00 ^a^ ± 1.73	160.92 ^b^ ± 6.02	26.23 ^b^ ± 2.08	3.44 ^b^ ± 0.33	4.08 ^b^ ± 0.55
**Double-layered films**	**3**	0.17 ^a^ ± 0.03	0.43 ^a^ ± 0.03	1.86 ^a^ ± 0.28	2.54 ^a^ ± 0.49	24.53 ^a^ ± 2.73	1.66 ^a^ ± 0.32	1.84 ^a^ ± 0.13	2.57 ^a^ ± 0.23
	**6**	0.66 ^a^ ± 0.05	0.90 ^a^ ± 0.27	3.66 ^a^ ± 0.91	51.35 ^a^ ± 9.85	52.36 ^a^ ± 2.98	9.01 ^a^ ± 0.10	2.37 ^a^ ± 0.39	2.74 ^a^ ± 0.38
	**10**	1.26 ^a^ ± 0.25	4.26 ^a^ ± 0.25	4.54 ^a^ ± 0.4	44.66 ^a^ ± 1.58	87.44 ^a^ ± 3.49	18.07 ^a^ ± 2.43	1.87 ^a^ ± 0.41	2.81 ^a^ ± 0.62

Values are presented as mean ± SD. TRPYP: tryptamine, PHEN: phenylethylamine, PUTRY: putrescine, CADAWE: cadawerin, HISTAM: histamine, TYRAM: tyramine, SPER: spermidine, and SPR: spermine. Films: samples stored in innovative films and LDPE: control samples stored in synthetic films. Different lettering next to the mean values for the LDPE and Double-layered films groups on individual days of storage indicate statistical differences between these groups (*p* < 0.01).

**Table 4 polymers-14-01717-t004:** Lipid oxidation of fish samples through storage in 4 °C.

Peroxide Value (Milliequivalents of Active Oxygen/kg Product)				
	Day 0	Day 3	Day 6	Day 10
**LDPE**	7.47 ± 0.34	14.58 ^a^ ± 0.36	35.18 ^a^ ± 0.59	46.10 ^a^ ± 0.15
**Double-layered films**		14.51 ^a^ ± 0.14	27.55 ^a^ ± 2.40	35.70 ^a^ ± 1.79
Acid value [mg KOH/g]				
**LDPE**	1.910.33	2.78 ^a^ ± 0.06	3.91 ^a^ ± 0.05	8.33 ^a^ ± 0.46
**Double-layered films**		2.21 ^a^ ± 0.29	3.83 ^a^ ± 1.11	4.05 ^a^ ± 1.38
		TBARS [mg/kg]		
**LDPE**	0.23 ± 0.02	0.18 ^a^ ± 0.03	0.80 ^a^ ± 0.05	0.34 ^a^ ± 0.12
**Double-layered films**		1.11 ^b^ ± 0.25	2.38 ^b^ ± 0.30	1.79 ^b^ ± 0.06

Values are presented as mean ± SD. Different lettering next to the mean values for the LDPE and Double-layered films groups on individual days of storage indicate statistical differences between these groups (*p* < 0.01).

## Data Availability

Not applicable.

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
