# Peer review of "Attempt to Extend the Shelf-Life of Fish Products by Means of Innovative Double-Layer Active Biodegradable Films"

_polymers, 2022, doi:10.3390/polym14091717_

Round 1
Reviewer 1 Report
REVIEW polymers-1686709
“Attempt to Extend the Shelf-Life of Fish Products by Means of Innovative Double-Layer Active BioDegradable Films”
By Joanna Tkaczewska, Ewelina Jamróz, Paulina Guzik and MichaÅ‚ Kopeć
The present manuscript provides valuable information about the application of double-layer biodegradable films as means for biopreservation of perishable fish products (salmon fillets). The Introduction part discusses the properties of different natural compounds of plant and animal origin used as biodegradable materials in biopackaging and food biopreservation. The M&M section is well arranged and the methods used in the study are well described. The results are well presented, the figures and tables are clear and informative, showing the trends and effects of biodegradable films as means reducing microbial spoilage and prolonging the shelf-life of treated salmon fillets. The conclusion is adequate, and the used references are sufficient and up-to-date.
I would recommend the authors to cite the following article concerning the application of edible films based on CMC and bioproducts:
Tumbarski, Y., Todorova, M., Topuzova, M., Georgieva, P., Ganeva, Z., Mihov, R., Yanakieva, V. Antifungal Activity of Carboxymethyl Cellulose Edible Films Enriched with Propolis Extracts and Their Role in Improvement of the Storage Life of Kashkaval Cheese. Current Research in Nutrition and Food Science Journal. 2021, 9(2): 487-499.
http://doi.org/10.12944/CRNFSJ.9.2.12
Author Response
Dear Editor,
We would like to thank you and the Reviewers for the time and effort invested in reading and correcting our manuscript in a thorough manner, and also for your comments, which led to improvement in the manuscript’s quality. We have revised the manuscript in light of the suggestions and comments. We hope that the revision has improved the paper to a level of satisfaction. Red color in the text indicates corrections or modifications to the manuscript. Detailed responses to specific comments/suggestions/queries are listed below.
Reviewer 1
Comment: I would recommend the authors to cite the following article concerning the application of edible films based on CMC and bioproducts: Tumbarski, Y., Todorova, M., Topuzova, M., Georgieva, P., Ganeva, Z., Mihov, R., Yanakieva, V. Antifungal Activity of Carboxymethyl Cellulose Edible Films Enriched with Propolis Extracts and Their Role in Improvement of the Storage Life of Kashkaval Cheese. Current Research in Nutrition and Food Science Journal. 2021, 9(2): 487-499.
Response: Additional discussion has been added in lines 403-409.
Reviewer 2
Comment: Some of the English must be revised
Response: The manuscript has been proof-edited by an English native speaker, professional translating and proof-editing company: Katarzyna Smith-Nowak (M.A. in Translation Studies, professional translator and proof-editor since 2003); Co. info: AmE Native Katarzyna Smith-Nowak, ul. Miejscowa 8, 30-499 Kraków, Phone: 0048 505 990 391; e-mail: english.native123@gmail.com).
The text has been extensively and carefully proof-edited to ensure proper cohesion, coherence and style of the manuscript, while maintaining its substantive content. The general readability for an English-speaking audience should now be better, the flow of the text ensured.
Below, please find a point-by-point list regarding the types of alterations/items checked in the text:
- First and foremost - stylistic editing;
- Eradication of repetitions and redundancies;
- Correction of grammatical number;
- Consequential use of Polish diacritic signs;
- Exchanging colloquial linguistic items for more formal ones;
- Maintaining consistency of AmE spelling;
- Correction of preposition usage;
- Correction of article usage;
- Maintaining linguistic consistency within the text;
- Corrections regarding parts of speech;
- Maintaining cohesion and coherence;
- Changes in word order;
- Checking subsequently added fragments.
Comment: L17 toxicity test for?
Response: Adequate corrections have been made.
Comment: L22 Was it due to reduced microbial growth.
Response: This information has been added to the ‘Abstract’ (in line 23).
Comment: L23-24 Recheck English
Response: This sentence has been corrected.
Comment: L 90-92 The statement is confusing.
Response: This sentence has been re-written.
Comment: L115 Extract or fruit?
Response: This part of the ‘Methodology’ section has been rewritten.
Comment: L121 at room..
Response: This has been adquately corrected.
Comment: L125-126 Recheck English
Response: The English has been corrected in these lines.
Comment: L170 Respiratory of ? Reader doesn’t understand why there is respiration.
Response: We have changed all statements containing these two words.
Comment: Add number of replications in all experiments
Response: This information has been added to the ‘Methodology’ section.
Comment: L251 Add more discussion and citation. The hydrophilicity of the starch-based materials highly affected the biodegradation behavior of the biodegradable films (https://doi.org/10.1016/j.fpsl.2022.100844).
Response: We have added more discussion on this.
Comment: Table 1 Please add SD values for all data.
Response: Adequate corrections have been made.
Comment: L257 What does this mean?
Response: It counted as losses after the control is deducted. This information has been added.
Comment: L280 Add more discussion and citation. Polysaccharides with hydrophilic cellulose derivatives readily absorbed water from the environment which also plasticized and caused swelling of polymers (https://doi.org/10.1002/app.45533).
Response: Appropriate corrections have been made.
Comment: L281 Why? Add more discussion and citation. Interaction between functional groups which eliminated free hydroxyl groups and hydrogen bonding decreased solubility of the films (https://doi.org/10.1016/j.fpsl.2021.100787).
Response: We have added more information.
Comment: Fig. 3 B The decrease, was it significant? Statistical analysis is needed.
Response: Fresh fish are products very susceptible to the development of microorganisms. Therefore, it is a standard practice that no statistical analyses are performed when comparing and assessing the microbiological quality of fresh fish products. For microbiological tests, 3 independent replicates (3 various fillets from different batches of fish) were used, which are characterized by very high differentiation in terms of the amount of microorganisms in the biological material. Thus, it is often the case that despite the occurrence of clear trends, the statistical analysis does not indicate statistical differences. Already minimal variation in the handling of the materials during their collection causes the initial number of microorganisms between different fillets to be potentially drastically different, which results in a large standard deviation. In a number of scientific publications on this topic found in reputable journals, the authors conclude that various factors affect the microbiological quality of fish products, without needing to conduct statistical analysis (https://doi.org/10.1016/j.fm.2017.10.011, https://doi.org/10.1016/j.foodchem.2008.09.078 https://doi.org/10.1016/j.foodcont.2022.108933 https://doi.org/10.1016/j.foodchem.2021.129347). This is the generally-accepted standard for the procedure. The authors of the work have published many works on the microbiological analysis of fish products (https://doi.org/10.1016/j.foodchem.2020.127867 https://doi.org/10.1016/j.foodchem.2022.132425)
Comment: Fig. 3 Add error bars (SD values).
Response: The standard division has been added.
Comment: L326 Films containing plant extract commonly had antimicrobial activity depending on the release behavior in to food and package atmosphere (https://doi.org/10.1016/j.lwt.2021.112356).
Response: Additional discussion has been added to lines 399-404.
Comment: L335 Remove space in front of ,
Response: This has been corrected.
Comment L425 Add more discussion and citation. Microbial growth caused modification of protein conformation including secondary structures and proteolysis which modified amino acid profiles in muscle foods (https://doi.org/10.1016/j.meatsci.2020.108367).
Response: Additional discussion has been added to lines 459-463.
Comment: L426 Recheck the use of capital letter for amino acid name. It may not be necessary.
Response: This has been corrected.
Comment: L456 Rate? The unit for rate should be as a function of time.
Response: The correction has been made.
Comment: Table 4 It is confusing. Unclear treatment between LLDPE and Films. LLDPE was not films? What film?
Response: The names of the study groups have been changed.
Comment: TBARS values: Add the statistical analysis
Response: Needed corrections have been made.
Comment: L475 Add more discussion and citation. Incorporations of active ingredients containing hydrophilic groups possibly reduced oxygen permeability, leading to lower oxygen transfer and lipid oxidation (https://doi.org/10.1016/j.fpsl.2020.100521).
Response: Additional discussion has been added in lines 530-532.
Reviewer 3.
Comment: The start of the abstract is not acceptable, and it is suggested to revise the sentence: “New, environmentally-friendly, biopolymer double-layer films with the addition of fish processing waste and active lingonberry extract were produced”.
Response: This part of the ‘Abstract’ has been reworded.
Comment: In the abstract describe some methodologies in brief and mention the most important numerical results of this study.
Response: Additional information has been added.
Comment: What are the advantages of using CSGH and lingberry in the previous studies?
Response: This information has been added to lines 93-97.
Comment: The manuscript contains many typo errors such as line 118 “and the of the total polyphenols”. Or line 120: “Uusing carboxymethyl”. Please correct the whole of the manuscript.
Response: These errors/typos have been corrected.
Comment: In the methodology section, the brand and model of all the instruments should be mentioned. Such as stirrer, blender, dryer, furnace, oven, incubator, etc.
Response: This information has been added.
Coment: Line 135: the writing style of this sentence is not an academic style: “Following digestion of the ash in diluted (1:2) HNO3, we transferred the samples to flasks”
Response: The proper corrections have been made.
Comment: Figure 1 is showing two figures, if they are the same minimize them to one figure, if there is a difference between them, discuss and give different captions to the figures.
Response: The correction has been made.
Comment: The authors can use the following reference in this study:
Sabbagh, F., Muhamad, I. I., PaLe, N., & Hashim, Z. (2018). Strategies in Improving Properties of Cellulose-Based Hydrogels for Smart Applications. Cellulose-Based Superabsorbent Hydrogels. Springer International Publishing, 887-908.
Response: Thank you for your suggestion. We have added this reference in lines 42-47.
Comment: Some references are too old (before 2010), it is recommended to change them with some newly published papers.
Response: This has been changed in accordance with the suggestions.

Reviewer 2 Report
The manuscript provide data for novel materials based on biopolymer. However, there are several points to be reconsidered. Significant of the statistical analysis is missing. Some of the English must be revised.
L17 toxicity test for?
L22 Was it due to reduced microbial growth
L23-24 Recheck English
L90-92 The statement is confusing.
L115 Extract or fruit?
L121 at room..
L125-126 Recheck English
L170 Respiratory of ? Reader doesn’t understand why there is respiration.
Add number of replications in all experiments.
L251 Add more discussion and citation. The hydrophilicity of the starch-based materials highly affected the biodegradation behavior of the biodegradable films (https://doi.org/10.1016/j.fpsl.2022.100844).
Table 1 Please add SD values for all data.
L257 What does this mean?
L280 Add more discussion and citation. Polysaccharides with hydrophilic cellulose derivatives readily absorbed water from the environment which also plasticized and caused swelling of polymers (https://doi.org/10.1002/app.45533).
L281 Why? Add more discussion and citation. Interaction between functional groups which eliminated free hydroxyl groups and hydrogen bonding decreased solubility of the films (https://doi.org/10.1016/j.fpsl.2021.100787).
Fig. 3 B The decrease, was it significant? Statistical analysis is needed.
Fig. 3 Add error bars (SD values).
L326 Films containing plant extract commonly had antimicrobial activity depending on the release behavior in to food and package atmosphere (https://doi.org/10.1016/j.lwt.2021.112356).
L335 Remove space in front of ,
L425 Add more discussion and citation. Microbial growth caused modification of protein conformation including secondary structures and proteolysis which modified amino acid profiles in muscle foods (https://doi.org/10.1016/j.meatsci.2020.108367).
L426 Recheck the use of capital letter for amino acid name. It may not be necessary.
L456 Rate? The unit for rate should be as a function of time.
Table 4 It is confusing. Unclear treatment between LLDPE and Films. LLDPE was not films? What film?
TBARS values: Add the statistical analysis
L475 Add more discussion and citation. Incorporations of active ingredients containing hydrophilic groups possibly reduced oxygen permeability, leading to lower oxygen transfer and lipid oxidation (https://doi.org/10.1016/j.fpsl.2020.100521).
Author Response
Dear Editor,
We would like to thank you and the Reviewers for the time and effort invested in reading and correcting our manuscript in a thorough manner, and also for your comments, which led to improvement in the manuscript’s quality. We have revised the manuscript in light of the suggestions and comments. We hope that the revision has improved the paper to a level of satisfaction. Red color in the text indicates corrections or modifications to the manuscript. Detailed responses to specific comments/suggestions/queries are listed below.
Reviewer 1
Comment: I would recommend the authors to cite the following article concerning the application of edible films based on CMC and bioproducts: Tumbarski, Y., Todorova, M., Topuzova, M., Georgieva, P., Ganeva, Z., Mihov, R., Yanakieva, V. Antifungal Activity of Carboxymethyl Cellulose Edible Films Enriched with Propolis Extracts and Their Role in Improvement of the Storage Life of Kashkaval Cheese. Current Research in Nutrition and Food Science Journal. 2021, 9(2): 487-499.
Response: Additional discussion has been added in lines 403-409.
Reviewer 2
Comment: Some of the English must be revised
Response: The manuscript has been proof-edited by an English native speaker, professional translating and proof-editing company: Katarzyna Smith-Nowak (M.A. in Translation Studies, professional translator and proof-editor since 2003); Co. info: AmE Native Katarzyna Smith-Nowak, ul. Miejscowa 8, 30-499 Kraków, Phone: 0048 505 990 391; e-mail: english.native123@gmail.com).
The text has been extensively and carefully proof-edited to ensure proper cohesion, coherence and style of the manuscript, while maintaining its substantive content. The general readability for an English-speaking audience should now be better, the flow of the text ensured.
Below, please find a point-by-point list regarding the types of alterations/items checked in the text:
- First and foremost - stylistic editing;
- Eradication of repetitions and redundancies;
- Correction of grammatical number;
- Consequential use of Polish diacritic signs;
- Exchanging colloquial linguistic items for more formal ones;
- Maintaining consistency of AmE spelling;
- Correction of preposition usage;
- Correction of article usage;
- Maintaining linguistic consistency within the text;
- Corrections regarding parts of speech;
- Maintaining cohesion and coherence;
- Changes in word order;
- Checking subsequently added fragments.
Comment: L17 toxicity test for?
Response: Adequate corrections have been made.
Comment: L22 Was it due to reduced microbial growth.
Response: This information has been added to the ‘Abstract’ (in line 23).
Comment: L23-24 Recheck English
Response: This sentence has been corrected.
Comment: L 90-92 The statement is confusing.
Response: This sentence has been re-written.
Comment: L115 Extract or fruit?
Response: This part of the ‘Methodology’ section has been rewritten.
Comment: L121 at room..
Response: This has been adquately corrected.
Comment: L125-126 Recheck English
Response: The English has been corrected in these lines.
Comment: L170 Respiratory of ? Reader doesn’t understand why there is respiration.
Response: We have changed all statements containing these two words.
Comment: Add number of replications in all experiments
Response: This information has been added to the ‘Methodology’ section.
Comment: L251 Add more discussion and citation. The hydrophilicity of the starch-based materials highly affected the biodegradation behavior of the biodegradable films (https://doi.org/10.1016/j.fpsl.2022.100844).
Response: We have added more discussion on this.
Comment: Table 1 Please add SD values for all data.
Response: Adequate corrections have been made.
Comment: L257 What does this mean?
Response: It counted as losses after the control is deducted. This information has been added.
Comment: L280 Add more discussion and citation. Polysaccharides with hydrophilic cellulose derivatives readily absorbed water from the environment which also plasticized and caused swelling of polymers (https://doi.org/10.1002/app.45533).
Response: Appropriate corrections have been made.
Comment: L281 Why? Add more discussion and citation. Interaction between functional groups which eliminated free hydroxyl groups and hydrogen bonding decreased solubility of the films (https://doi.org/10.1016/j.fpsl.2021.100787).
Response: We have added more information.
Comment: Fig. 3 B The decrease, was it significant? Statistical analysis is needed.
Response: Fresh fish are products very susceptible to the development of microorganisms. Therefore, it is a standard practice that no statistical analyses are performed when comparing and assessing the microbiological quality of fresh fish products. For microbiological tests, 3 independent replicates (3 various fillets from different batches of fish) were used, which are characterized by very high differentiation in terms of the amount of microorganisms in the biological material. Thus, it is often the case that despite the occurrence of clear trends, the statistical analysis does not indicate statistical differences. Already minimal variation in the handling of the materials during their collection causes the initial number of microorganisms between different fillets to be potentially drastically different, which results in large standard deviation. In a number of scientific publications on this topic found in reputable journals, the authors conclude that various factors affect the microbiological quality of fish products, without needing to conduct statistical analysis (https://doi.org/10.1016/j.fm.2017.10.011, https://doi.org/10.1016/j.foodchem.2008.09.078 https://doi.org/10.1016/j.foodcont.2022.108933 https://doi.org/10.1016/j.foodchem.2021.129347). This is the generally-accepted standard for the procedure. The authors of the work have published many works on the microbiological analysis of fish products (https://doi.org/10.1016/j.foodchem.2020.127867 https://doi.org/10.1016/j.foodchem.2022.132425)
Comment: Fig. 3 Add error bars (SD values).
Response: The standard division has been added.
Comment: L326 Films containing plant extract commonly had antimicrobial activity depending on the release behavior in to food and package atmosphere (https://doi.org/10.1016/j.lwt.2021.112356).
Response: Additional discussion has been added to lines 399-404.
Comment: L335 Remove space in front of ,
Response: This has been corrected.
Comment L425 Add more discussion and citation. Microbial growth caused modification of protein conformation including secondary structures and proteolysis which modified amino acid profiles in muscle foods (https://doi.org/10.1016/j.meatsci.2020.108367).
Response: Additional discussion has been added to lines 459-463.
Comment: L426 Recheck the use of capital letter for amino acid name. It may not be necessary.
Response: This has been corrected.
Comment: L456 Rate? The unit for rate should be as a function of time.
Response: The correction has been made.
Comment: Table 4 It is confusing. Unclear treatment between LLDPE and Films. LLDPE was not films? What film?
Response: The names of the study groups have been changed.
Comment: TBARS values: Add the statistical analysis
Response: Needed corrections have been made.
Comment: L475 Add more discussion and citation. Incorporations of active ingredients containing hydrophilic groups possibly reduced oxygen permeability, leading to lower oxygen transfer and lipid oxidation (https://doi.org/10.1016/j.fpsl.2020.100521).
Response: Additional discussion has been added in lines 530-532.
Reviewer 3.
Comment: The start of the abstract is not acceptable, and it is suggested to revise the sentence: “New, environmentally-friendly, biopolymer double-layer films with the addition of fish processing waste and active lingonberry extract were produced”.
Response: This part of the ‘Abstract’ has been reworded.
Comment: In the abstract describe some methodologies in brief and mention the most important numerical results of this study.
Response: Additional information has been added.
Comment: What are the advantages of using CSGH and lingberry in the previous studies?
Response: This information has been added to lines 93-97.
Comment: The manuscript contains many typo errors such as line 118 “and the of the total polyphenols”. Or line 120: “Uusing carboxymethyl”. Please correct the whole of the manuscript.
Response: These errors/typos have been corrected.
Comment: In the methodology section, the brand and model of all the instruments should be mentioned. Such as stirrer, blender, dryer, furnace, oven, incubator, etc.
Response: This information has been added.
Coment: Line 135: the writing style of this sentence is not an academic style: “Following digestion of the ash in diluted (1:2) HNO3, we transferred the samples to flasks”
Response: The proper corrections have been made.
Comment: Figure 1 is showing two figures, if they are the same minimize them to one figure, if there is a difference between them, discuss and give different captions to the figures.
Response: The correction has been made.
Comment: The authors can use the following reference in this study:
Sabbagh, F., Muhamad, I. I., PaLe, N., & Hashim, Z. (2018). Strategies in Improving Properties of Cellulose-Based Hydrogels for Smart Applications. Cellulose-Based Superabsorbent Hydrogels. Springer International Publishing, 887-908.
Response: Thank you for your suggestion. We have added this reference in lines 42-47.
Comment: Some references are too old (before 2010), it is recommended to change them with some newly published papers.
Response: This has been changed in accordance with the suggestions.

Reviewer 3 Report
- The start of the abstract is not acceptable, and it is suggested to revise the sentence: “New, environmentally-friendly, biopolymer double-layer films with the addition of fish processing waste and active lingonberry extract were produced”.
- In the abstract describe some methodologies in brief and mention the most important numerical results of this study.
- What are the advantages of using CSGH and lingberry in the previous studies?
- The manuscript contains many typo errors such as line 118 “and the of the total polyphenols”. Or line 120: “Uusing carboxymethyl”. Please correct the whole of the manuscript.
- In the methodology section, the brand and model of all the instruments should be mentioned. Such as stirrer, blender, dryer, furnace, oven, incubator, etc.
- Line 135: the writing style of this sentence is not an academic style: “Following digestion of the ash in diluted (1:2) HNO3, we transferred the samples to flasks”
- Figure 1 is showing two figures, if they are the same minimize them to one figure, if there is a difference between them, discuss and give different captions to the figures.
- The authors can use the following reference in this study:
Sabbagh, F., Muhamad, I. I., PaLe, N., & Hashim, Z. (2018). Strategies in Improving Properties of Cellulose-Based Hydrogels for Smart Applications. Cellulose-Based Superabsorbent Hydrogels. Springer International Publishing, 887-908.
- Some references are too old (before 2010), it is recommended to change them with some newly published papers.
- Just as a suggestion, the authors can add a section as “Future Prospects” at the end of the manuscript.
Author Response

(The authors gave the same response as above.)

Round 2
Reviewer 2 Report
The manuscript has been revised as recommend.